# Electrochemical Properties of Multilayered Sn/TiNi Shape-Memory-Alloy Thin-Film Electrodes for High-Performance Anodes in Li-Ion Batteries

**DOI:** 10.3390/ma15072665

**Published:** 2022-04-05

**Authors:** Duck-Hyeon Seo, Jun-Seok Lee, Sang-Du Yun, Jeong-Hyeon Yang, Sun-Chul Huh, Yon-Mo Sung, Hyo-Min Jeong, Jung-Pil Noh

**Affiliations:** 1Department of Energy & Mechanical Engineering and Institute of Marine Industry, Gyeongsang National University, 2 Tongyeonghaean-daero, Tongyeong 53064, Korea; dudug17@gnu.ac.kr (D.-H.S.); no9192@gnu.ac.kr (J.-S.L.); schuh@gnu.ac.kr (S.-C.H.); ysung@gnu.ac.kr (Y.-M.S.); hmjeong@gnu.ac.kr (H.-M.J.); 2Department of Mechanical System Engineering, Gyeongsang National University, 2 Tongyeonghaean-daero, Tongyeong 53064, Korea; sdyun@gnu.ac.kr (S.-D.Y.); jh.yagi@gnu.ac.kr (J.-H.Y.)

**Keywords:** Sn anode, TiNi shape memory alloy, stress-absorption effect, multilayer thin film

## Abstract

Sn is a promising candidate anode material with a high theoretical capacity (994 mAh/g). However, the drastic structural changes of Sn particles caused by their pulverization and aggregation during charge–discharge cycling reduce their capacity over time. To overcome this, a TiNi shape memory alloy (SMA) was introduced as a buffer matrix. Sn/TiNi SMA multilayer thin films were deposited on Cu foil using a DC magnetron sputtering system. When the TiNi alloy was employed at the bottom of a Sn thin film, it did not adequately buffer the volume changes and internal stress of Sn, and stress absorption was not evident. However, an electrode with an additional top layer of room-temperature-deposition TiNi (TiNi(RT)) lost capacity much more slowly than the Sn or Sn/TiNi electrodes, retaining 50% capacity up to 40 cycles. Moreover, the charge-transfer resistance decreased from 318.1 Ω after one cycle to 246.1 Ω after twenty. The improved cycle performance indicates that the TiNi(RT) and TiNi-alloy thin films overall held the Sn thin film. The structure was changed so that Li and Sn reacted well; the stress-absorption effect was observed in the TiNi SMA thin films.

## 1. Introduction

Rechargeable Li-ion batteries (LIBs) are widely used in portable electronics, electric vehicles, and energy storage systems [1]. However, to meet the rising electric-power demand, new LIBs must be developed with high capacity and long cycle performance [2]. Current commercial graphite anodes have an unsuitably low theoretical capacity of 372 mAh/g [3,4]. Therefore, other anode materials with higher capacity such as Si, Sn, SnO_2_, and Li have attracted interest [5,6,7,8,9,10,11,12,13,14,15,16].

Sn is a promising material for metal-based anodes; it is economical and has a high theoretical capacity of 994 mAh/g and a moderate operation potential [6]. However, as illustrated in Figure 1a, the lager volume expansion and contraction of Sn particles during lithiation–delithiation causes pulverization and cracking, and thus, fading capacity. To overcome this problem and improve the cycle life, volume change and internal stress need to be suppressed or alleviated [7,8,9].

Recently, many attempts have been made to minimize the larger volume change and internal stress of the Sn anode. Zhang et al. manufactured a Sn–Ni–Cu alloy by electrodeposition and subsequent heat treatment, which enhanced electroconductivity and effectively buffered the volume change of the Sn anode [10]. Polat et al. reported that the multilayer cycle performance of a Ni–Sn–C anode was enhanced by Ni and C, which also increased resistance against volume expansion [11]. Multilayer Sn/graphitized-carbon fabricated by Chang et al. showed enhanced cycle stability because of the structural solidity of graphitized carbon [12].

TiNi shape-memory alloy (SMA) has also been reported to inhibit volume expansion and absorb stress. Joo et al. demonstrated that Si thin film deposited on a TiNi-alloy thin film improved cycle performance through the stress-absorption effect of TiNi SMA. The problems of Sn anodes are similar to those of Si anodes; thus, it would be expected that the volume expansion and internal stress of Sn anodes could also be minimized by TiNi SMA [13].

In this study, TiNi SMA was employed as a buffer matrix to suppress the volume change and internal stress of a Sn anode. The stress-absorption effect was caused by the superelasticity, in turn caused by the stress-induced martensitic transformation of TiNi SMA [17,18]. When the B2 phase of the TiNi SMA is stressed, it is converted into a B19′-martensite, but it changes back when the stress is removed. The internal stress of the Sn anode is then able to convert the TiNi SMA phase to appear superelastic. Hu et al. reported that the expansion stress generated when Sn is converted into Li_2_Sn_5_ is approximately 2.91 GPa, whereas the critical stress of the B2–B19′ transition at room temperature is 35.5 MPa; therefore, the stress generated from the volume expansion of the Sn anode may cause the martensite phase conversion [19]. As shown in Figure 1b, if TiNi SMA is introduced into a Sn anode, volume expansion and stress are generated during lithiation, but the TiNi SMA absorbs the stress and changes from B2 to B19′-martensite. Later, stress is removed because of volume contraction during delithiation, and the martensite returns to the B2 phase. Therefore, the structure of the Sn anode may be maintained, and its electrochemical properties improved.

Sn/TiNi SMA multilayer thin films were fabricated in this study using a DC magnetron sputtering system. The effect of the superelasticity of the TiNi SMA on the electrochemical performance of the Sn anode was investigated through electrochemical and material analysis. Superelastic TiNi SMAs can accommodate the volume changes and stress occurring in the Sn electrode during lithiation–dilithiation.

## 2. Experimental

### 2.1. Production of Sn/TiNi Multilayer Thin Films

Sn/TiNi multilayer thin films were deposited on Cu foil by a DC magnetron sputtering system using pure Sn (99.99%), Ti (99.99%), and Ni (99.99%) targets. The target-substrate distance was 80 mm. Before deposition of the thin film, the surface of each Cu foil was washed with acetone and alcohol. The base and working pressures in the chamber were 5.0 × 10^−6^ and 2.0 × 10^−3^ torr, respectively. The Ar gas-flow rate was 20 sccm. The surface of each target was pre-sputtered for 15 min to eliminate impurities. First, the TiNi-alloy thin film was deposited with Ti power of 180 W and Ni power of 60 W for 120 min on Cu foil at 580 °C. Then, the Sn thin film was deposited with Sn power of 50 W for 15 min at room temperature. Finally, the TiNi-alloy thin film was deposited with the same Ti and Ni power conditions as above for 60 min at room temperature (the TiNi-alloy thin film deposited at room temperature will be called TiNi(RT)) Sn, Sn/TiNi, and TiNi(RT)/Sn/TiNi thin films were produced.

### 2.2. Materials Characterization

The morphology and composition of the thin films were observed through a field-emission scanning electron microscope (FE-SEM, Mira3 LM, TESCAN, Kohoutovice Czech Republic) with an energy-dispersive spectrometer (EDS) at 15 kV. The crystallographic structure was determined with an X-ray diffractometer (XRD, Ultima IV, Rigaku, Tokyo, Japan) with Cu K_α_ radiation.

### 2.3. Electrochemical Characterization

For the electrochemical test, coin cells (CR2032) were assembled in an Ar-filled glove box. A specific area (1 cm^2^) of the sample was used as the working electrode. The thin films were used directly as binder-free working electrodes without conductive additives. The counter electrode and electrolyte used were, respectively, Li foil and 1 M LiPF_6_ in a mixture of ethylene carbonate and diethyl carbonate (DEC) (1:1 vol%). A constant-current (CC) charge–discharge test was performed with a current of 180 μA (0.5C) in a voltage window from 0.01 to 1.2 V (vs. Li^+^/Li). Cyclic voltammetry (CV) was conducted at a scan rate of 0.1 mV/s in a voltage window from 0.01 to 2.0 V (vs. Li^+^/Li). Electrochemical impedance spectroscopy was performed at an amplitude of 10 mV in the frequency range of 1 MHz~10 mHz.

## 3. Results and Discussion

The surface, cross-section, and composition of the deposited Sn/TiNi thin films were investigated by FE-SEM and EDS. Figure 2a shows the surface of the Cu foil substrate. Figure 2b shows the surface image of a TiNi-alloy thin film deposited on Cu foil. The surface morphology was very rough: the TiNi grain size was approximately 180 nm. As shown in Figure 2c, the Sn thin film deposited on the TiNi-alloy thin film was also rough; the grain size of Sn was approximately 780 nm. Figure 2d shows a cross-sectional image of Sn/TiNi deposited on a glass slide. The thicknesses of the Sn and TiNi thin films were 0.5 µm and 2.5 µm, respectively. Figure 3 presents the EDS results for TiNi thin film: Ti-49.91 at.% and Ni-50.09 at.%. Ti and Ni were uniformly distributed on the surface.

The crystal structures of the deposited TiNi and Sn/TiNi thin films were analyzed by XRD (Figure 4). The diffraction peak of the (110) plane of the B2 phase was observed at 42.2° in the XRD pattern of TiNi. In that of Sn/TiNi, the diffraction peak of the B2 phase was also at 42.2°; additional diffraction peaks at 30.4°, 31.8°, 43.8°, 44.7°, 55.1°, 62.5°, 63.6°, and 64.5° were assigned to the (200), (101), (220), (211), (301), (112), (400), and (321) planes, respectively, of tetragonal Sn (JCPDS-#862265). Sn and B2-phase TiNi alloys were successfully deposited as there were no impurity peaks in the XRD pattern.

The electrochemical reactions of the Sn and Sn/TiNi electrodes were examined by CV. Figure 5a,b, respectively, show the CV-test results of the Sn and Sn/TiNi electrodes evaluated at a scan rate of 0.1 mV/s in the voltage range 0.01–2.0 V. The lower-right insets of Figure 5a,b are the corresponding first-cycle CV-test results. In the lower-right inset of Figure 5a, there was a reduction peak at 1.23 V during discharge; this corresponded to the formation of a solid-electrolyte interface (SEI) [20]. The SEI was formed in the first-cycle discharge, which led to a large irreversible capacity loss. The second-cycle-discharge reduction peaks at 0.65, 0.4, 0.28, and 0.01 V may be attributed to the alloying of Li-Sn; the second-cycle-charge oxidation peaks at 0.5, 0.59, 0.71, and 0.79 V correspond to the dealloying of Li_x_Sn [21]. The CV graphs of the second and third cycles overlapped well, which indicates high reversibility. The CV results with the Sn/TiNi electrode (Figure 5b) resembled those with the Sn electrode, indicating that the TiNi-alloy thin film had little effect on the electrochemical reaction at the Sn anode.

The electrochemical properties of the Sn and Sn/TiNi electrodes were investigated by CC discharge–charge testing. Figure 6a,b exhibits the discharge–charge curves of the Sn and Sn/TiNi electrodes at 0.5 C in the range of 0.01 V–1.2 V, respectively. Plateaus occurred at certain potentials during charging and discharging; these were attributed to alloying/dealloying of Li-Sn and corresponded to the potential-peaks in the CV results for the Sn electrode. The first-cycle discharge capacities of the Sn and Sn/TiNi electrodes were 493 and 506 mAh/g, respectively; the second-cycle-discharge capacities were 442 and 447 mAh/g. The difference was due to the large irreversible capacity loss caused by SEI formation.

Figure 7 shows the cycle performance of the Sn and Sn/TiNi electrodes. The capacity of the Sn electrode was constant up to 10 cycles, but rapidly degraded thereafter. Until the 19th cycle, 50% capacity was maintained. The rapid degradation of capacity was the result of pulverization and cracking due to the rapid volume expansion and internal stress of the Sn electrode during charging and discharging. The Sn/TiNi electrode sustained constant capacity for five cycles; this declined to 50% capacity by the seventh cycle. The high rate of capacity degradation only slowed significantly after the ninth cycle. Despite the introduction of the TiNi alloy, the cycle performance did not improve compared to that of the Sn electrode. When the TiNi-alloy thin film was introduced into the lower part of the Sn thin film, it evidently failed to accommodate the volume expansion and internal stress of the Sn, and superelasticity did not appear.

To accommodate the volume expansion and internal stress of Sn in the Sn/TiNi electrodes more effectively, a different TiNi-alloy thin film—TiNi(RT)—was deposited on the Sn thin film. However, Sn has a melting point of 232 °C and a temperature of 580 °C when depositing the TiNi-alloy thin film, so the upper TiNi-alloy thin film was deposited at room temperature. The composition of the TiNi(RT) alloy thin film was the same as that of the TiNi-alloy thin film according to the EDS results. In general, TiNi alloys deposited at room temperature are well known to be amorphous [22] and not to exhibit superelasticity; the new film was thus only intended to suppress volume expansion of Sn. The total TiNi(RT)/Sn/TiNi thin film was manufactured as an electrode and its electrochemical properties were analyzed.

Figure 8 shows the CV-test results for the TiNi(RT)/Sn/TiNi electrode. There were four reduction peaks at 0.66, 0.33, 0.12, and 0.01 V and four oxidation peaks at 0.51, 0.67, 0.75, and 0.81 V, similar to those of the Sn electrode. The four reduction peaks were related to the alloying of Li-Sn, and the four oxide peaks to dealloying. Electrochemical reactions occurred even when the TiNi(RT)-alloy thin film that did not react with Li was deposited on the Sn thin film, as shown in Figure 8. It is possible that the TiNi(RT) alloy does not interfere with the diffusion of Li ions into the active Sn material. Therefore, the TiNi(RT)-alloy thin film does not affect the electrochemical reaction of Sn.

The electrochemical performance of the TiNi(RT)/Sn/TiNi electrode was measured by CC testing. Figure 9 shows the discharge–charge curves. Plateaus appeared at certain potentials during the charge–discharge process in the TiNi(RT)/Sn/TiNi electrode, at locations almost identical to those of the peaks in the CV graph. These plateaus may be attributed to the alloying/dealloying of Li-Sn. The first-cycle discharge capacity of the TiNi(RT)/Sn/TiNi electrode was 497 mAh/g.

Figure 10 compares the cycle performance of the Sn, Sn/TiNi, and TiNi(RT)/Sn/TiNi electrodes. The TiNi(RT)/Sn/TiNi electrode maintained constant capacity from one to 10 cycles, after which the capacity decreased slowly, unlike those of the Sn and Sn/TiNi electrodes. In addition, 50% capacity was still maintained by TiNi(RT)/Sn/TiNi at the 40th cycle; this performance was approximately twice as good as that of the Sn electrode. The cycle performance improved because the TiNi(RT) and TiNi-alloy thin films effectively suppressed the volume expansion generated during charge and discharge, and the internal stress was absorbed because of the superelasticity of the TiNi-alloy thin film. Thus, the electrode structure was maintained.

To find the cause of the cycle-performance improvement in the TiNi(RT)/Sn/TiNi electrode, electrochemical impedance spectroscopy was performed at a voltage amplitude of 10 mV in a frequency range of 1 MHz to 0.01 Hz. Figure 11a shows the Nyquist impedance graphs for the Sn, Sn/TiNi, and TiNi(RT)/Sn/TiNi electrodes after first-cycle discharge. The diameter of the semicircle indicates the charge-transfer resistance (R_ct_). For Sn, Sn/TiNi, and TiNi(RT)/Sn/TiNi electrodes, the R_ct_ values after first-cycle discharge were 321, 333.9, and 318.1 Ω, respectively. Thus, the TiNi(RT) alloy thin film did not interrupt Li-ion diffusion as the R_ct_ of the TiNi(RT)/Sn/TiNi electrode was similar to those of the Sn and Sn/TiNi electrodes. Figure 11b shows the impedance graphs of the electrodes after 20th-cycle discharge. The R_ct_ values of the Sn, Sn/TiNi, and TiNi (RT)/Sn/TiNi thin-film electrodes were 339.1, 305.83, and 246.13 Ω, respectively. After the 20th-cycle discharge in the TiNi(RT)/Sn/TiNi electrode, the R_ct_ decreased more than that after the first-cycle discharge because the electrode structure of Sn was maintained to some extent. This confirms the superelasticity of the TiNi-alloy and TiNi(RT)-alloy thin films, and shows that Li and Sn were changed to react well. Li ions were better transferred during cycles, improving the cycle performance.

In this study, the cycle increase effect was obtained by utilizing the stress absorption effect of the TiNi thin film as opposed to using the existing Cu current collector. In future, performance improvements may be possible by experimenting with the thickness ratio of the Sn and TiNi thin films to achieve maximum stress absorption efficiency, and by depositing a crystalline TiNi thin film on the upper layer.

## 4. Conclusions

In this study, TiNi SMA was introduced into a Sn thin film as a buffer matrix. Sn-TiNi multilayer thin films were manufactured using a DC magnetron sputtering system. The Sn/TiNi thin-film electrode did not improve the cycle performance compared to the Sn thin-film electrode. When a TiNi alloy was introduced only at the bottom of a Sn thin film, it did not accommodate the volume expansion and internal stress of Sn well, and the effect of the superelasticity was not clearly apparent. A TiNi(RT)/Sn/TiNi electrode underwent slower capacity degradation than the Sn or Sn/TiNi electrodes and maintained at least 50% capacity up to 40 cycles. Furthermore, the R_ct_ of TiNi(RT)/Sn/TiNi electrode decreased from 318.1 Ω after one cycle to 246.1 Ω after 20 cycles. The cycle performance improved because the TiNi(RT) and TiNi-alloy thin films effectively buffered the volume expansion of Sn, and because Li and Sn were transformed to react well. The stress-absorption effect appeared in the TiNi SMA thin films.

## Figures and Tables

**Figure 1 materials-15-02665-f001:**
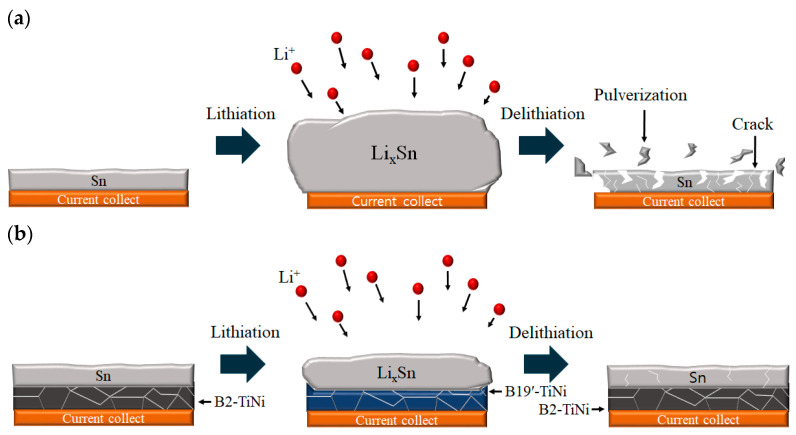
Schematic of the volume-change mechanism in (**a**) Sn anodes; (**b**) Sn-TiNi thin-film anodes during charge/discharge cycling.

**Figure 2 materials-15-02665-f002:**
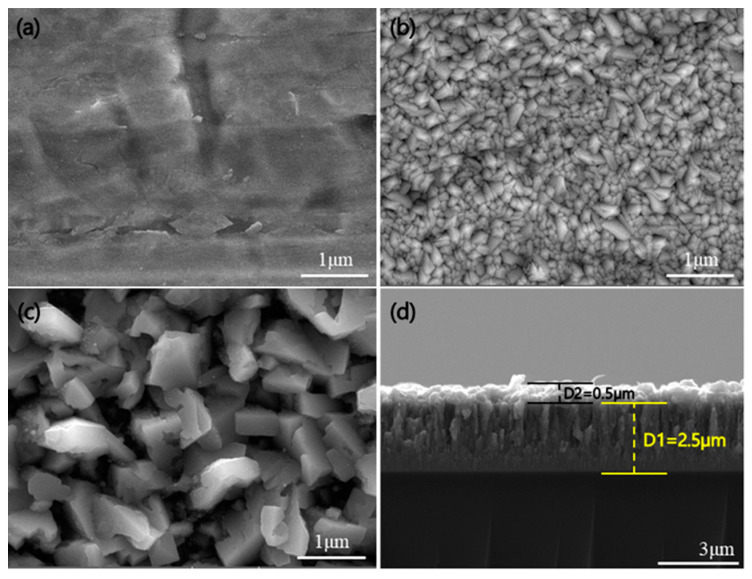
FE-SEM surface images of (**a**) Cu foil, (**b**) TiNi thin film, and (**c**) Sn thin film. (**d**) Cross-section of the Sn/TiNi thin film deposited on a glass substrate under the same growth conditions.

**Figure 3 materials-15-02665-f003:**
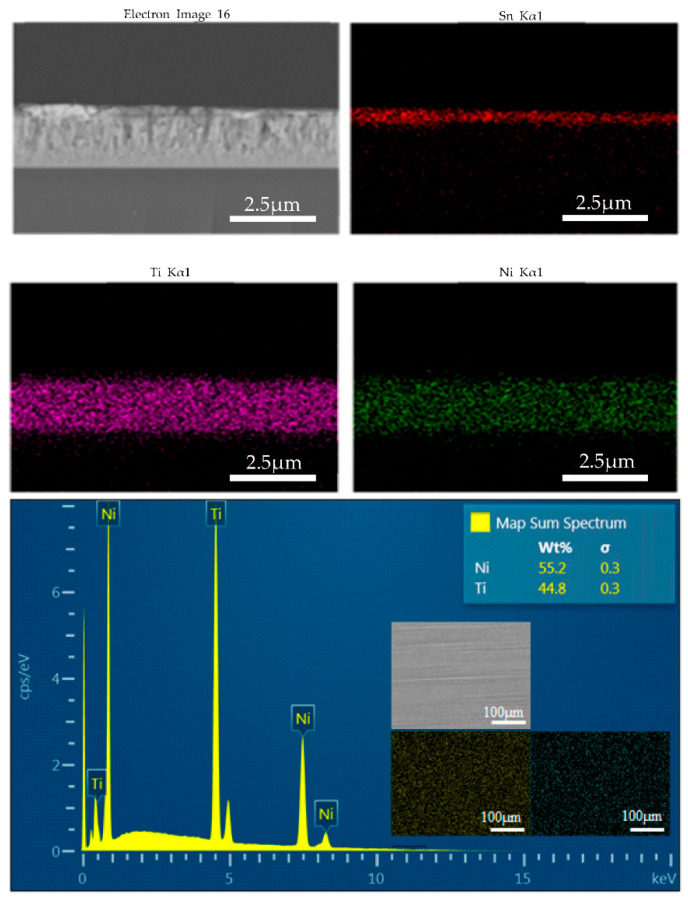
Energy-dispersive X-ray spectrum and mapping image of the TiNi thin film.

**Figure 4 materials-15-02665-f004:**
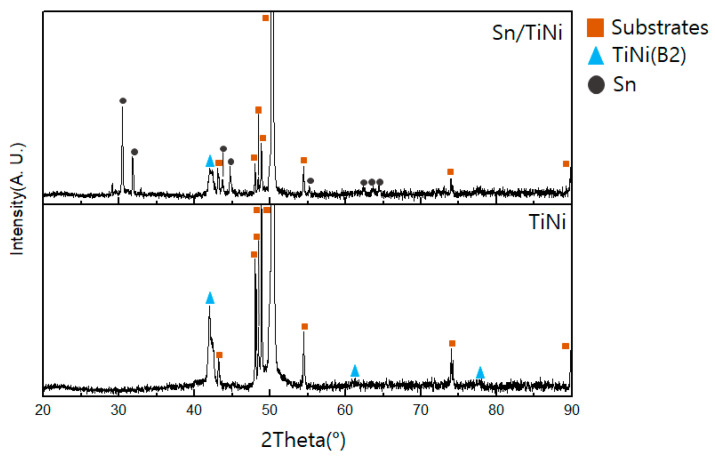
X-ray diffraction patterns of TiNi and Sn/TiNi thin films.

**Figure 5 materials-15-02665-f005:**
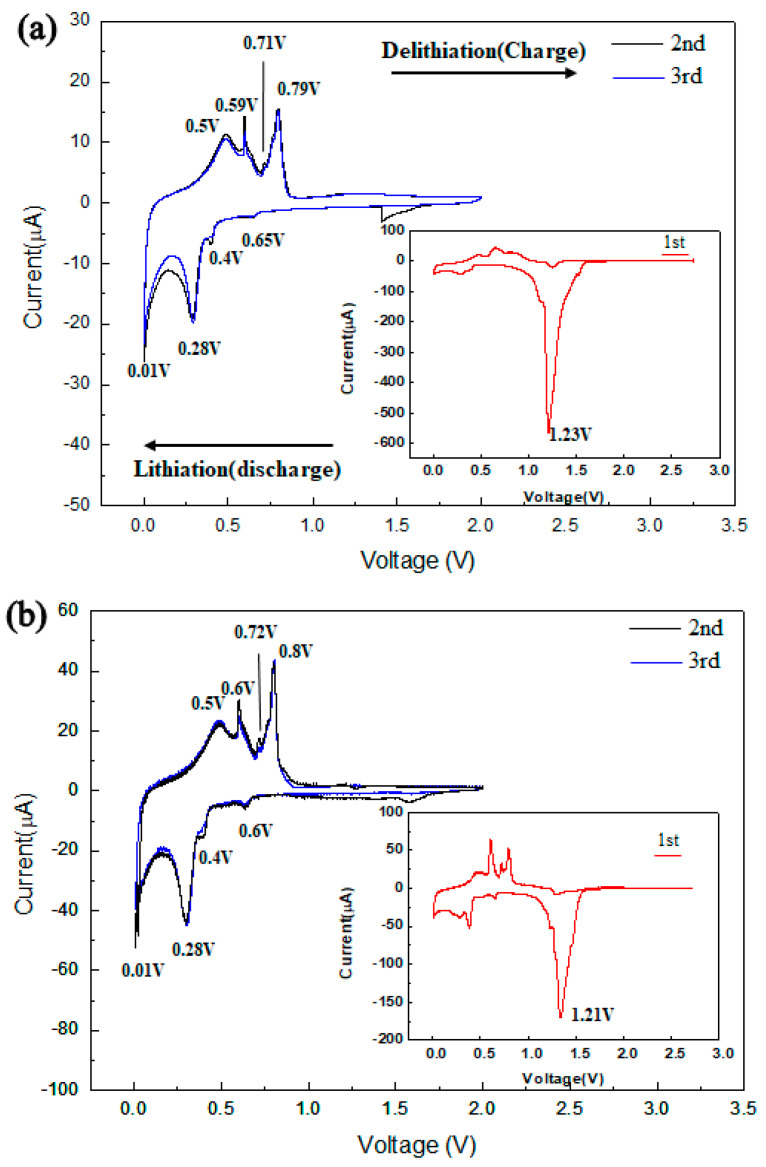
Cyclic voltammetry results for the (**a**) Sn and (**b**) Sn/TiNi electrode in the second and third cycles at 0.1 mV/s in the voltage window of 0.01~2.0 V (vs. Li^+^/Li). Insets show the results of the first cycle.

**Figure 6 materials-15-02665-f006:**
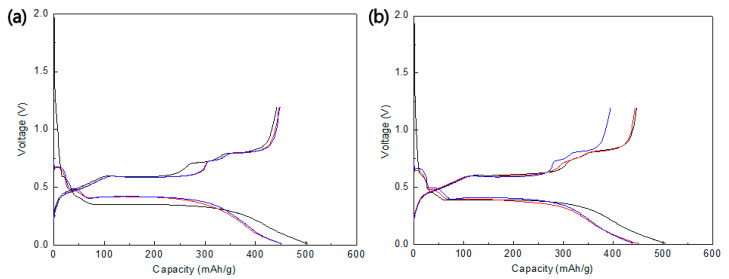
Discharge–charge curve of the (**a**) Sn and (**b**) Sn/TiNi electrode at 0.5 C in the voltage window of 0.01~1.2 V (vs. Li^+^/Li).

**Figure 7 materials-15-02665-f007:**
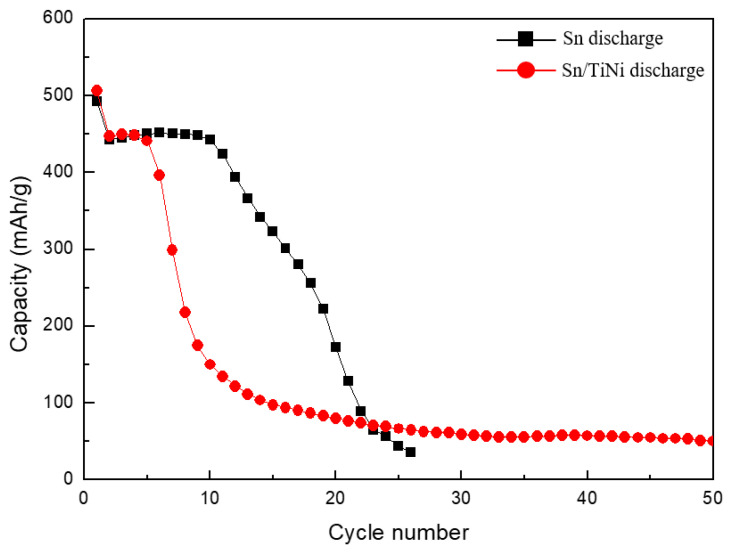
Cycle performance of the Sn and Sn/TiNi electrodes at 0.5 C.

**Figure 8 materials-15-02665-f008:**
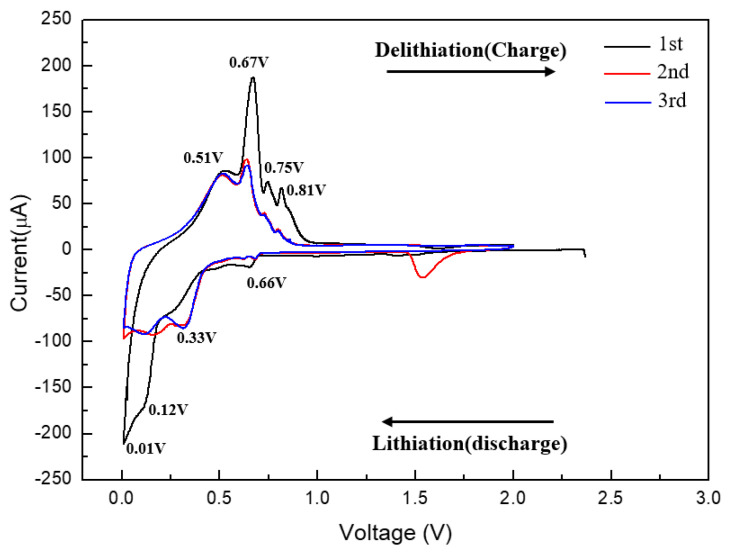
Cyclic voltammetry of the TiNi(RT)/Sn/TiNi electrode at 0.1 mV/s in the voltage window of 0.01~2.0 V (vs. Li^+^/Li).

**Figure 9 materials-15-02665-f009:**
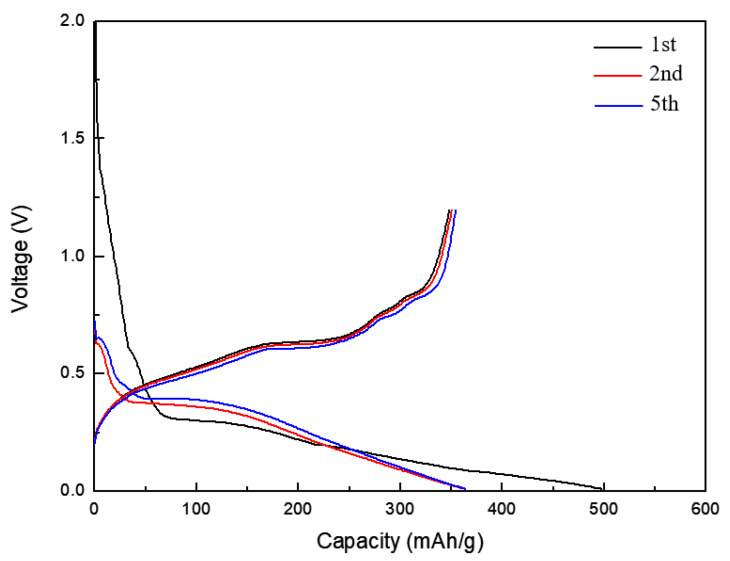
Discharge–charge curve of the TiNi(RT)/Sn/TiNi electrode at 0.5 C in the voltage window of 0.01~1.2 V (vs. Li^+^/Li).

**Figure 10 materials-15-02665-f010:**
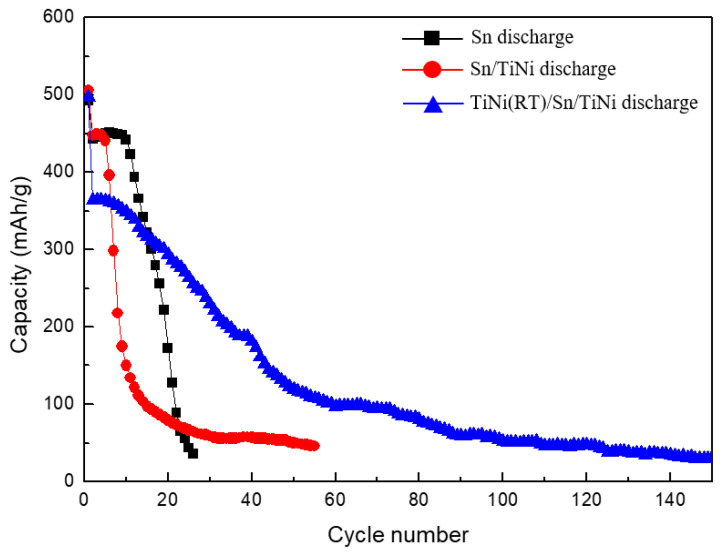
Cycle performance of Sn, Sn/TiNi, and TiNi(RT)/Sn/TiNi electrodes at 0.5 C.

**Figure 11 materials-15-02665-f011:**
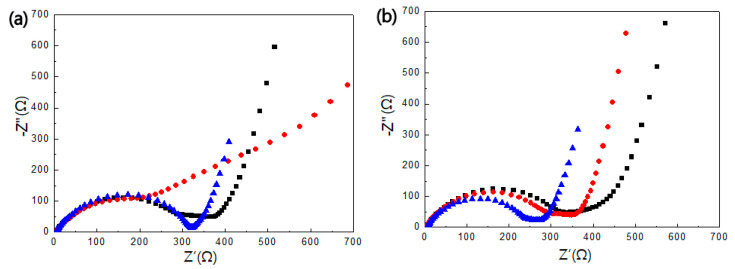
Nyquist impedance plots of the Sn, Sn/TiNi, and TiNi(RT)/Sn/TiNi electrodes after (**a**) first discharge and (**b**) 20th discharge at 0.5 C, measured in the frequency range from 0.01 Hz to 1 MHz. Z’: real part of impedance; Z’’: imaginary part of impedance.

## Data Availability

Not applicable.

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
