# Peer review of "Electrochemical Properties of Multilayered Sn/TiNi Shape-Memory-Alloy Thin-Film Electrodes for High-Performance Anodes in Li-Ion Batteries"

_materials, 2022, doi:10.3390/ma15072665_

Round 1

Reviewer 1 Report

Seo et al investigate electrochemical properties of multilayered Sn/TiNi shape-memory-alloy thin-film electrodes for high-performance Anodes in Li-ion batteries. However, the following modifications are required before the paper can be considered for acceptance. 

1. Sample size needs to be provided.

2. Sample morphology (SEM images) after electrochemical cycling should be provided.  The thickness variation of the cross section before and after the electrochemical cycling of the film needs to be provided.

3. The diversity of material characterization methods needs to be increased. EDS line scanning towards cross sections of the film is required.

4. The electrochemical performance of the sample is not high, the author should discuss the further improvement of its performance at the end of the paper.

5. Please discuss the possible advantages of material composition, structure, synthesis process and cost detailedly.

6. Some references are old. Some recently reported works (in the past three years) toward Sn-related materials are suggested to cite in the revision.
1) Rare Metals 2020, 39, 1159–1164. doi: 10.1007/s12598-014-0303-6.
2) Journal of Colloid and Interface Science 2021, 602, 563-572. doi:10.1016/j.jcis.2021.06.046.
3) Journal of Alloys and Compounds 2022, 896, 163009. doi: 10.1016/j.jallcom.2021.163009.

Author Response

Dear reviewer #1

We would like to submit a revised manuscript of our paper titled “Electrochemical properties of multilayered Sn/TiNi shape-memory-alloy thin-film electrodes for high-performance anodes in Li-ion batteries” (Manuscipt ID: materials-1654544). We have modified the manuscript accordingly, and the detailed answers are listed below point by point.

Reviewer 2 Report

In this work, the authors reported a multilayered TiNi(RT)/Sn/TiNi hybrid anode for Li ion battery by DC magnetron sputtering system. The TiNi SMA shows superelasticity, which can accommodate the volume change and internal stress of Sn anode. The resulted TiNi(RT)/Sn/TiNi anode delivers much stable electrochemical performance compare with the raw Sn anode. This paper could be accepted for publication in Materials after revision with the consideration of following questions:

  1. In Figure 2, the thickness of TiNi and Sn films are 2.5μm and 0.5μm. The capacity contribution of this hybrid anode is Sn, the TiNi act as a buffer layer which doesn’t show capacity in Li ion battery. The thick TiNi layer dramatically decrease the gravimetric capacity of entire Sn/TiNi hybrid anode. In addition, the current of 0.5 C-rate used in this work is only 180μA, the theoretical capacity of Sn is 994 mAh/g, so, that the mass loading of Sn in this work is only 0.3622 mg/cm2. Why did the authors deposit such thick TiNi layer and such thin Sn layer?
  2. In Figure 5, the authors claimed that the CV results of Sn/TiNi electrode resemble with Sn electrode. However, the Sn electrode shows obvious anodic and cathodic peaks, but the Sn/TiNi doesn’t has obvious peaks. The cathodic peak at 0.28 V disappears in Sn/TiNi anode. In addition, The CV curves of Sn/TiNi anode shows obvious wrong points, especially at the second cycle. The authors should provide a reasonable explanation.
  3. In the insets of Figure 5, both of the Sn and Sn/TiNi anodes show large irreversible capacity loss at the first cycle, which means that both of them should display low initial Coulombic efficiency at first discharge/charge process and large capacity decay after the first cycle. However, Figure 6 shows that both Sn and Sn/TiNi anodes show a high initial Coulombic efficiency of higher than 85%. The discharge capacity at the second cycle of both Sn and Sn/TiNi anodes is not far less than that of the first cycle, which is not consistent with the results of CV curves. In addition, the Coulombic efficiency of Sn/TiNi anode at the fifth cycle is far less than that of raw Sn anode. The authors should give reasonable explanations.
  4. Why does the TiNi(RT)/Sn/TiNi anode show lower capacity than that of Sn and Sn/TiNi anodes after the first cycle?

Author Response

Dear reviewer #2

We would like to submit a revised manuscript of our paper titled “Electrochemical properties of multilayered Sn/TiNi shape-memory-alloy thin-film electrodes for high-performance anodes in Li-ion batteries” (Manuscipt ID: materials-1654544). We have modified the manuscript accordingly, and the detailed answers are listed below point by point.

Round 2

Reviewer 1 Report

The author addressed most of the reviewer's concerns, except question 2. 

The authors state“owing to problems separating the active material from the current collector, the experiment on the morphology of the sample after cycling failed. ”

In fact, the authors did not need to separate the active material from the current collector at all. That's exactly the advantage of this monolithic electrode. Considering that this detection plays an important supporting role in this paper, it is suggested that the author overcome the difficulties and complete the following characterization:

Sample morphology (SEM images) after electrochemical cycling should be provided. The thickness variation of the cross section before and after the electrochemical cycling of the film needs to be provided.

Author Response

We resubmit a revised manuscript “Electrochemical properties of multilayered Sn/TiNi shape-memory-alloy thin-film electrodes for high-performance anodes in Li-ion batteries” (Manuscipt ID: materials-1654544).

Reviewer #1

The author addressed most of the reviewer's concerns, except question 2. 

The authors state“owing to problems separating the active material from the current collector, the experiment on the morphology of the sample after cycling failed. ”

In fact, the authors did not need to separate the active material from the current collector at all. That's exactly the advantage of this monolithic electrode. Considering that this detection plays an important supporting role in this paper, it is suggested that the author overcome the difficulties and complete the following characterization:

Sample morphology (SEM images) after electrochemical cycling should be provided. The thickness variation of the cross section before and after the electrochemical cycling of the film needs to be provided.

à Thanks for the reviewer’s good comments.

Sn has a disadvantage in large volume expansion during lithiation and delithiation. Therefore, we think that the rapid deterioration of the cycle is also due to the destruction of the Sn thin film. After the cycle was finished, the Sn thin film was decomposed and could not maintain the thin film shape. So, when the coin cell was disassembled, the shape of the sample could not be maintained.

In the future, we will disassemble the sample in the middle of the cycle and try to observe the surface and cross section.

As pointed out by the reviewer, we will continue research and report it in the next paper.

Reviewer 2 Report

The authors corrected this paper appropriatelly and answered all the qurstions. This paper can be published on Materials.

Author Response

Dear reviewer #2

We resubmit a revised manuscript “Electrochemical properties of multilayered Sn/TiNi shape-memory-alloy thin-film electrodes for high-performance anodes in Li-ion batteries” (Manuscipt ID: materials-1654544).

Reviewer #2

The authors corrected this paper appropriatelly and answered all the qurstions. This paper can be published on Materials.

à Thanks for the reviewer’s good comments.
